# Response of a Sylvan Moss Species (*Didymodon validus* Limpr.) with a Narrow Distribution Range to Climate Change

**Tingting Wu, Chuntong Pan, Tao Bian, Qiaoxin Wang, Jin Kou \* and Bangwei Zhou \***

Key Laboratory of Vegetation Ecology, Ministry of Education, School of Life Sciences,
Northeast Normal University, Changchun 130024, China; wutt488@nenu.edu.cn (T.W.);
panchuntong@nenu.edu.cn (C.P.); wangqiaoxin@nenu.edu.cn (Q.W.)
\*   Correspondence: kouj398@nenu.edu.cn (J.K.); zhoubw599@nenu.edu.cn (B.Z.)

**Abstract:** Mosses are particularly susceptible to climate change owing to their close biological and ecological associations with climatic conditions. However, there is a limited understanding of the changes in distribution patterns of the moss species in forest ecosystems under climate change, especially in mosses with narrow ranges. Therefore, we reconstructed historical, simulated present, and predicted future potential distribution patterns of *Didymodon validus*, a narrow-range moss species in the forest ecosystem, using the MaxEnt model. The aim of this study was to explore its unique suitable habitat preference, the key environmental factors affecting its distribution, and the distributional changes of *D. validus* under climate change at a long spatial-time scale. Our findings indicate that the most suitable locations for *D. validus* are situated in high-altitude regions of southwestern China. Elevation and mean temperature in the wettest quarter were identified as key factors influencing *D. validus* distribution patterns. Our predictions showed that despite the dramatic climatic and spatial changes over a long period of time, the range of *D. validus* was not radically altered. From the Last Interglacial (LIG) to the future, the area of the highly suitable habitat of *D. validus* accounted for only 15.3%–16.4% of the total area, and there were weak dynamic differences in *D. validus* at different climate stages. Under the same climate scenarios, the area loss of suitable habitat is mainly concentrated in the northern and eastern parts of the current habitat, while it may increase in the southern and eastern margins. In future climate scenarios, the distribution core zone of suitable habitat will shift to the southwest for a short distance. Even under the conditions of future climate warming, this species may still exist both in the arid and humid regions of the QTP in China. In summary, *D. validus* showed cold and drought resistance. Our study provides important insights and support for understanding the impact of climate change on the distribution of *D. validus*, as well as its future distribution and protection strategies.

**Keywords:** *Didymodon validus*; MaxEnt; paleoclimate; climate warming; shifts

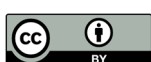

## 1. Introduction

Bryophytes, including mosses, liverworts, and hornworts, are a group of non-vascular plants that encompass over 20,000 recorded species worldwide. They inhabit diverse terrestrial habitats, ranging from tropical to polar regions and the sub-Antarctic [1,2]. Several studies have demonstrated the significant role of mosses in carbon and nutrient cycling processes [3,4], permafrost stability [4], water retention, pedogenesis [5], colonization by higher plants [6], and ecological restoration efforts [7]. Furthermore, due to their unistratose leaves and lack of well-developed cuticles, bryophytes have no resistance to ion exchange; thus, they are highly sensitive to changes in the surrounding environment [5]. Furthermore, bryophytes are more exposed to the effects of climate than vascular plants because their physiology is strongly linked to climatic factors, such as temperature and precipitation [8]. For example, it has been reported that bryophytes will increase,

migrate, or become extinct due to climate change [9–11]. However, a limited number of studies have explored the response of key bryophyte taxa to climate change.

*Didymodon* is one of the largest genera in the Pottiaceae family, including about 140 species distributed worldwide, and it occurs in temperate areas, especially mountainous and drought-prone regions [12–16]. It has been reported that *Didymodon* species are the primary components of biological soil crusts in arid and semi-arid areas, with important ecological functions [17,18]. Furthermore, due to a strong sensitivity to climate change, *Didymodon* is recommended as a climate indicator [13,16]. Although *Didymodon* species generally have strong adaptability to extreme climate and environmental conditions, some individual species often face more survival challenges due to their unique habitats and adaptions.

*Didymodon validus* Limpr. is a sylvan moss species with a relatively narrow distribution range. It was described as a new species by Limpricht (1888) and has been reported to be distributed intermittently in Europe (Austria, Germany, Italy, Switzerland, Slovakia), Central (Kirghizstan) and Southwest Asia, the Arabian Peninsula, and Ethiopia (Tigray province) [19,20]. However, this species has been known in China for a relatively short time. Shuayib et al. (2017) first found *D. validus* on the Tomur Peak and the Altun Mountains in Xinjiang, China. Subsequently, *D. validus* samples were collected in forest regions in Yunnan and Xizang through our fieldwork. A previous study showed that climate change greatly affects the distribution of narrow-range species, which may be at risk of extinction [21]. However, there has been a lack of study on the distribution pattern, origin, and shifts of *D. validus* as climate change in different timelines.

The species distribution models (SDMs), also called ecological or environmental niche models, have a pivotal function in quantifying species–habitat relationships and projecting species distributions in ecological research, conservation, and environmental management [22–24]. Typical SDMs include MaxEnt, GARP, ENFA, and Bioclim [22,24–26]. Due to the advantages of high accuracy, simple operation, and the small size of the sample, MaxEnt has proven to be a top-performing algorithm in comparison to other methods [22,23,27,28]. The MaxEnt model has been widely applied to simulate the geographical distribution of specified species in the past [29], current [30], and future [24], which have also been utilized to predict the geographical distribution, including didymodon on regional scals [13,16,31]. However, no broad-scale studies have predicted the distribution patterns of *Didymodon* at the species level.

Under these circumstances, we utilized the MaxEnt model to simulate the distribution pattern of *D. validus* across China under different climatic scenarios. The objectives of this study include (1) revealing its unique suitable habitat preference for *D. validus*, (2) determining the relationship between the distribution of *D. validus* and climatic and topographic factors, and (3) exploring the geographical distribution changes of *D. validus* under different climatic conditions at a large spatial-time scale (from LIG to 2070). This study will enable the ongoing conservation of bryophytes and their habitats.

## 2. Materials and Methods

### 2.1. Species Traits

Because of previous confusion over the taxonomy and distribution of *D. validus* [32], we only considered data from within China in the present study. During our continuous investigation of xerophytic moss, particularly *Didymodon* Hedw., in different provinces in China, some *D. validus* specimens were collected in Tibet and Yunnan. Microscopic examinations and measurements were taken with a ZEISS Primo Star light microscope (ZEISS, Oberkochen, Germany). Morphological observations were obtained with a Canon EOS 70D camera (Minato Ward, Tokyo, Japan) mounted on a microscope. Specimens were examined in 2% KOH [33].

## 2.2. Study Area and Species Occurrence

We obtained occurrence data for *D. validus* from field investigations, documented literature, and checked herbarium information. The specific field investigation can be found in our previous study [15,16,33,34]. Recently, *D. validus* was collected and identified by us and other domestic researchers in the process of studying the genus *Didymodon*. Data and systematic survey results across China [13,16,31,35–37] indicate that its distribution range is limited to Xizang, Shilin, Yunnan, Tomur Peak in Tianshan, and Altun Mountain, Xinjiang. The environments in these areas were generally characterized by high altitude, strong solar radiation, low annual temperature variance, and high daily temperature variance [13,38,39]. Based on accurate species identification and distribution records, 8 distribution points of *D. validus* were used in our analysis (Figure 1). The national fundamental geographic data were provided by the National Geomatics Center of China (http://www.cehui8.com/3S/GIS/20130702/205.html (accessed on 5 November 2020)).

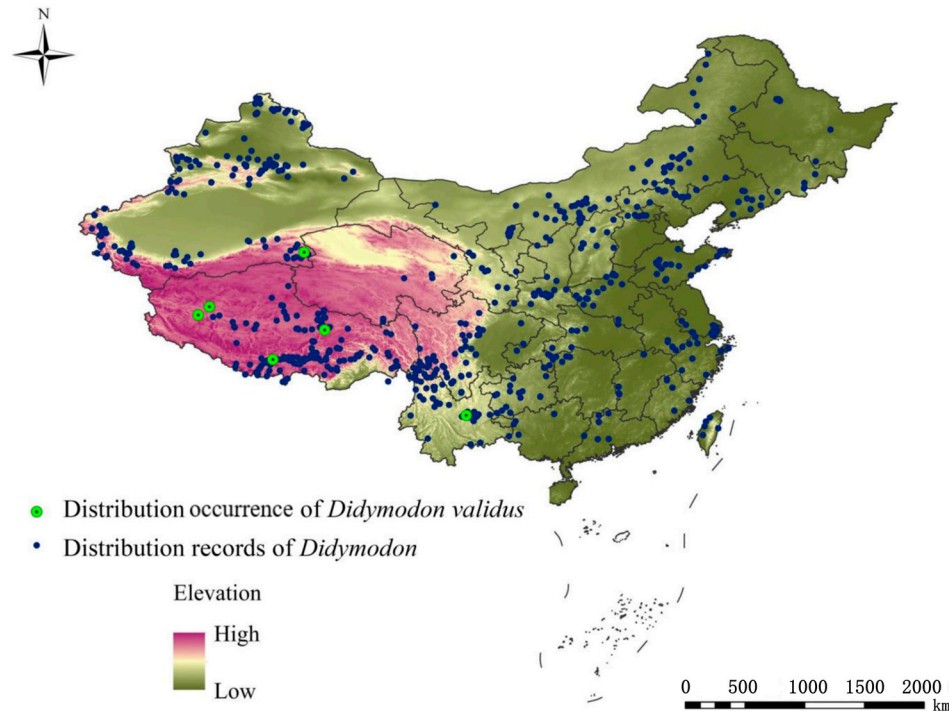

**Figure 1.** Distribution occurrence of *D. validus* in China.

## 2.3. Environmental Variables and Climate Change Scenarios

Considering the importance of climatic and topographic data in determining past, present, and future distribution of *D. validus* in China, we used climatic data downloaded from the WorldClim database (http://www.worldclim.org/ (accessed on 5 June 2020)) from three paleoclimate datasets: Last Interglacial period (LIG, about 120–140 ka, a warm and humid climate), Last Glacial Maximum (LGM, about 22 ka, the climate became colder and drier), and Mid Holocene (MH, about 6 ka, the climate warmed up) in the Community Climate System Model four (CCSM4) global climate model; one present dataset: the Current (1970–2000); four Representative Concentration Pathways (RCPs) climate change data, where 2050 and 2070 use average emissions for the years 2041 to 2060 and 2061 to 2080, respectively. We used combinations of RCP 2.6–2050, RCP 2.6–2070, RCP 8.5–2050, and RCP 8.5–2070 with CCSM4 to simulate global climate responses to increased greenhouse gas emissions [40,41]. For the topographic variables, digital elevation model data were obtained from the USGS GTOPO 30 series (https://www1.gsi.go.jp/geowww/global-map-gsi/gtopo30/gtopo30.html (accessed on 5 June 2020)) and then used to derive the

aspect and slope data using ArcGIS 10.5 (Esri, Redlands, CA, USA). The climate data were used to run the model-covered areas in China. All environmental layers were resampled to a 2.5 min resolution in ArcGIS 10.5. To deal with collinearity, we performed variance inflation factors (VIFs) and Pearson correlation analyses. As a result, six bioclimatic variables (Bio2 (mean diurnal range), Bio8 (mean temperature of wettest quarter), Bio9 (mean temperature of driest quarter), Bio13 (precipitation of wettest month), Bio15 (precipitation seasonality), Bio19 (precipitation of coldest quarter)), and three topographic variables (elevation, slope, and aspect) were selected.

### 2.4. Distribution Modeling

The MaxEnt model (version 3.4.1, http://biodiversityinformatics.amnh.org/open_source/maxent/ (accessed on 14 July 2020)) was used to predict the probability of the distribution of *D. validus* across China [42]. A total of 80% of the occurrence data were used for model training and 20% for model testing to estimate the model's capacity. The recommended default values were used for the convergence threshold ($10^{-5}$), with the maximum iterations (500) and the maximum number of background points (10,000) using the recommended default parameters (i.e., regularization multiplier = 1). The selection of environmental variables and functions was carried out automatically under the default rules, and 10 random partitions were created in the occurrence data [43]. Furthermore, the jackknife approaches and response curves were plotted to demonstrate how the variables affected the potential species distribution [44]. We reclassified potentially suitable habitats into four categories: high (0.6–1), moderate (0.4–0.6), low (0.2–0.4), and none (0–0.2) [45]. We projected the fitted models onto past, current, and future climate conditions. All binary distribution projections were stacked from the different climate conditions to explore central tendencies in projections, and we selected overlapping areas among projections as a future distribution range [46]. The range change and core zone shifts were predicted by comparing the binary outputs in different periods above using a Python-based GIS toolkit SDM toolbox [47].

### 2.5. Model Evaluation

To further evaluate the performance of the modeling algorithms, the area under the curve (AUC) and the true skill statistic (TSS) were used [48,49]. The value of the AUC ranges between 0.5 and 1, with >0.9 representing excellent predictive performance of the model, 0.7–0.9 indicating moderately useful models, and 0.5 representing randomness [50,51]. The TSS ranges from −1 to 1, where −1 indicates a perfect inverse prediction, 1 indicates perfect performance, and 0 indicates randomness [48]. The threshold was set to the value at which the TSS was maximized ($TSS_{max}$). The R package Biomod2 was applied to conduct the TSS assessment [52]. Both the final values of AUC and TSS were produced by an average of 10 replicates.

## 3. Results

### 3.1. Morphological Characteristics

Plants, medium-sized, growing in dense turfs, brown below, green above. Stems, 1.5–2.1 cm high, erect to ascending, usually branched, weakly radiculose at base, in transverse section, rounded, hyalodermis absent, central strand differentiated, and sclerodermis present. Axillary hairs, filiform, long, usually 5–8 cells long, with one brown basal cell and hyaline upper ones. Rhizoidal tubers are absent. Leaves, twisted and incurved when dry, spreading when moist, 2–3 × 0.65–0.75 mm, channeled ventrally in the upper part, sheathing. Lamina is completely unistratose and yellow with KOH. Apex acuminate, not apiculate, not cucullate. Margins entire, recurved from base to 2/3 to 3/4 of the leaf, unistratose. Costa, 95–162.5 μm wide at the base, long-excurrent, and not spurred. Ventral cells of the costa in the upper middle part of the leaf quadrate or subquadrate, smooth or with low papillose. Dorsal cells of the costa in the upper middle part of the leaf quadrate or

subquadrate, smooth. Transverse section semicircular to elliptic, with 4–7 guide cells in one layer below midleaf, 1–3 layers of ventral stereids, and 2–4 layers of dorsal stereids, without hydroids. Ventral surface cells are differentiated, not bulging, and smooth. Dorsal surface cells are differentiated and smooth. Upper and middle laminal cells subquadrate or oblate, 7–9.5 × 7.75–9.75 μm, smooth and thick-walled. Basal cells are not differentiated, smooth, basal juxtacostal cells short-rectangular to rectangular, 20–37.5 × 8.75–13.75 μm. Basal marginal cells oblate or quadrate, 5.25–8.75 × 7.5–12.75 μm. Gemmae are absent. Sporophytes are unknown. The morphological characteristics are shown in Figure 2.

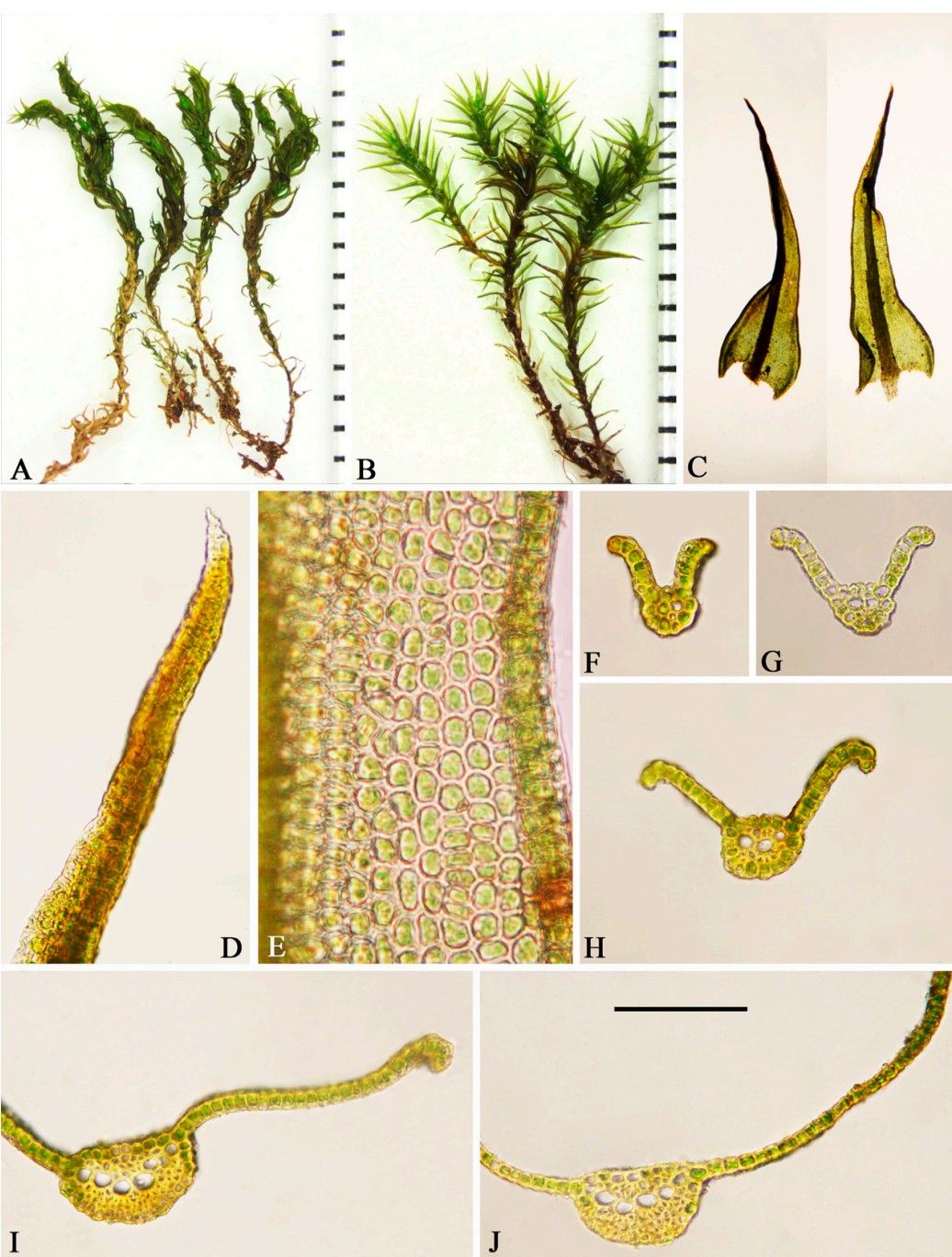

**Figure 2.** *Didymodon validus*. (**A**) Plants when dry (scale in mm). (**B**) Plants when moist (scale in mm). (**C**) Leaves. (**D**) Leaf apex (dorsal). (**E**) Upper laminal cells. (**F**–**J**) Transverse section of leaf, sequentially from apex to base. Scale bar on (**J**): (**C**)—1 mm; (**D**,**F**–**J**)—100 μm; (**E**)—40 μm.

### 3.2. Potential Distribution Pattern of D. validus

Models for *D. validus* with a cross-validation AUC of 0.751 and a TSS value of 0.526 indicated that the model results (Figure 3) could be considered satisfactory for predicting potential suitable habitats for *D. validus*. The potential distribution pattern of *D. validus* is primarily located in Tibet, southern and central Xinjiang, southern and northern Qinghai, northwestern Sichuan, southwestern Gansu, and a small part of northwestern Yunnan. Among those, the most suitable areas (distribution probability over 0.6) are concentrated in the Qinghai–Tibet Plateau (QTP) and adjacent areas of southwestern China, especially in the high mountain area of 4000–6000 m, with a distribution probability greater than 0.7. In general, these areas with a high probability of distribution were relatively small and severely fragmented.

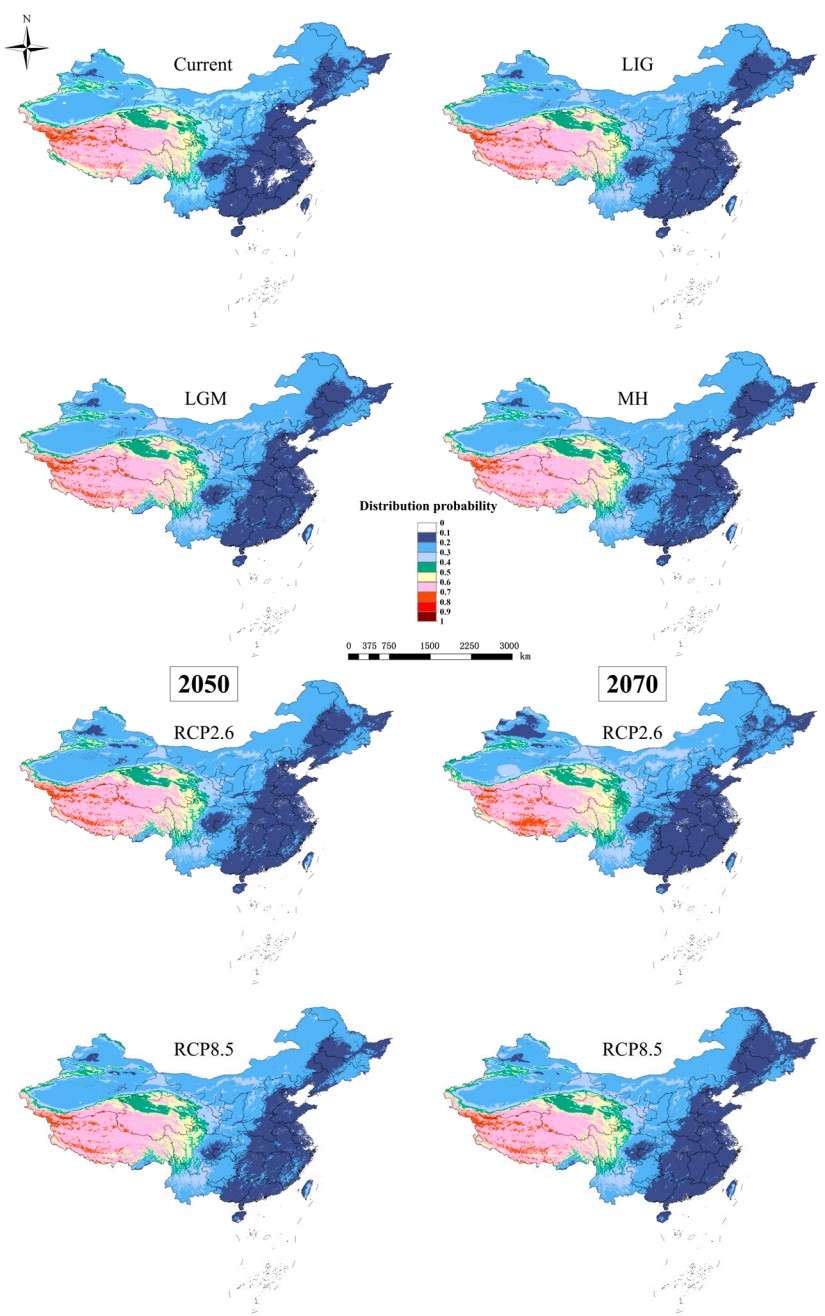

**Figure 3.** Potential suitable distribution areas for *D. validus* in China based on the MaxEnt model under different climatic conditions.

The proportions of areas within the four suitability classes of the potential distributions of *D. validus* in China are shown in Figure 4. For the different climate conditions, the area of the highly suitable habitat of *D. validus* accounted for only 15.3%–16.4% of the total area, indicating that the distribution of this species in China may be relatively narrow. In addition, there were weak dynamic differences in *D. validus* at different climate stages. In the historical climate period, we found that the proportion of highly suitable *D. validus* area in China was at a small increase stage and reached the highest value of 16.4% in the MH, which was reconstructed by the prediction model. In the current climate period, the highly suitable area of *D. validus* began to shrink, and the proportion of the highly suitable area reached its lowest value of 15.3% throughout the predicted climate period. Compared to the present distribution, the highly suitable habitat distribution range of the species was predicted to expand slightly in the future (2050s and 2070s), with the increase in suitable habitats in the RCP8.5 climate scenario being less than that in RCP2.6. In general, the potential distribution area of *D. validus* in China fluctuated little with climate change, and it had a relatively stable distribution and reproduction.

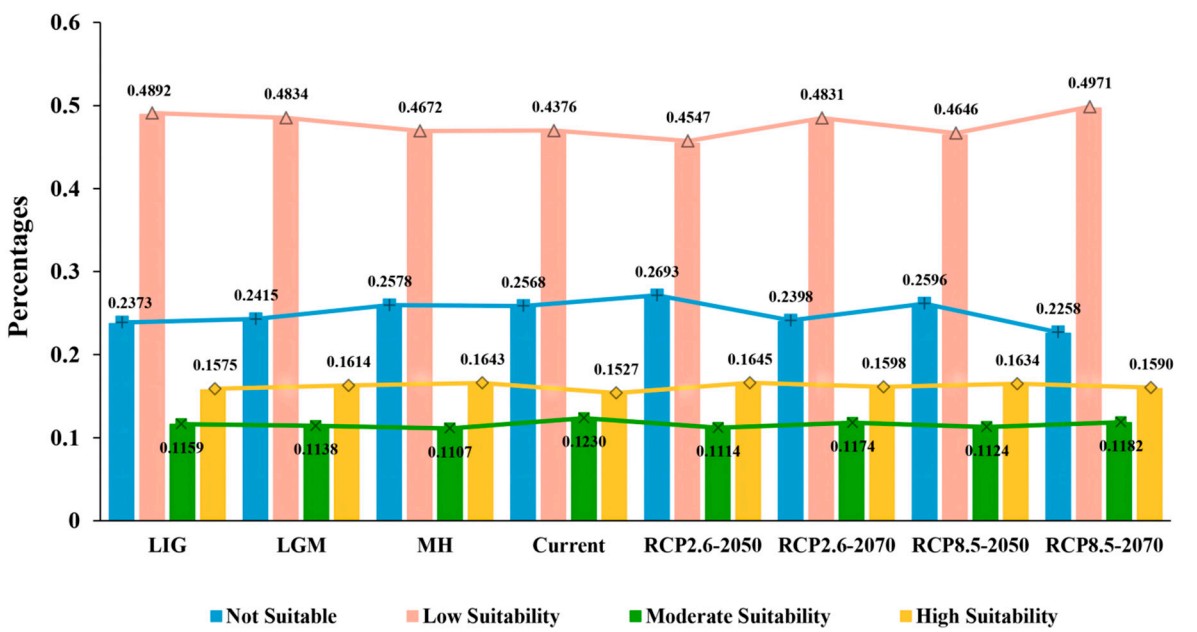

**Figure 4.** Proportions of areas within the four suitability classes of potential distributions of *D. validus* in China.

### 3.3. Relationship between Environmental Factors and the Distribution Pattern of D. validus

The importance of the relative contributions of the environmental variables considered in the modeling was evaluated using the jackknife test shown in Figure 5. The analysis revealed that the elevation exhibited the greatest gains in the MaxEnt model. The variables of mean temperature in the wettest quarter (Bio8) were important for shaping the *D. validus* distribution, and precipitation in the coldest quarter (Bio19) and precipitation in the wettest month (Bio13) were the main precipitation variables that influenced the potential distribution of *D. validus*. In contrast, the other variables exhibited relatively low gains, indicating that the weak contributions could almost be ignored.

## Jackknife of regularized training gain for *D. validus*

**Figure 5.** Jackknife test for evaluating the relative importance of environmental variables for distribution of *D. validus* in China.

The response curves of the nine variables for the habitat suitability of *D. validus* are shown in Figure 6. Based on the response curves of the major variables, the topography variables were stable environmental factors affecting the distribution of *D. validus*. It was predicted that *D. validus* mainly prefers habitats with high altitudes and large slopes in China, and within the predicted variable range, its survival probability gradually increased with the increase in elevation, reaching the maximum at about 6000 m altitude, indicating that *D. validus* is a typical alpine species in China. In addition, climatic variables were the main limiting factors for the potential distribution of *D. validus*. There were differences in the range of temperature and precipitation variables to which *D. validus* could adapt in different climatic periods. During the LIG period, *D. validus* seemed to have a wider climate adaptation. For example, when the mean temperature in the driest quarter (Bio9) was between −22 and 46 °C and the precipitation seasonality (Bio15) was from 30 to 240, *D. validus* may be present. However, from the MH period, the habitat adaptability of *D. validus* gradually tended to be stable. The occurrence probability was higher (over 0.5) when the mean temperature in the wettest quarter (Bio8) was −15–15 °C and precipitation in the coldest quarter (Bio19) was 0–20 mm. According to the response curves of the main variables, some precipitation variables and temperature variables were negatively correlated with the probability of *D. validus* distribution. For example, when the values of precipitation in the wettest month (Bio13) and precipitation in the coldest quarter (Bio19) increased, the probability of the presence of *D. validus* decreased. Similarly, as the value of the mean temperature in the wettest quarter (Bio8) decreased, the probability of the presence of *D. validus* increased. In general, *D. validus* is a typical alpine bryophyte adapted to cold and drought climates in China, with a relatively narrow range of optimum growth temperature and precipitation, and is mainly distributed at high elevations.

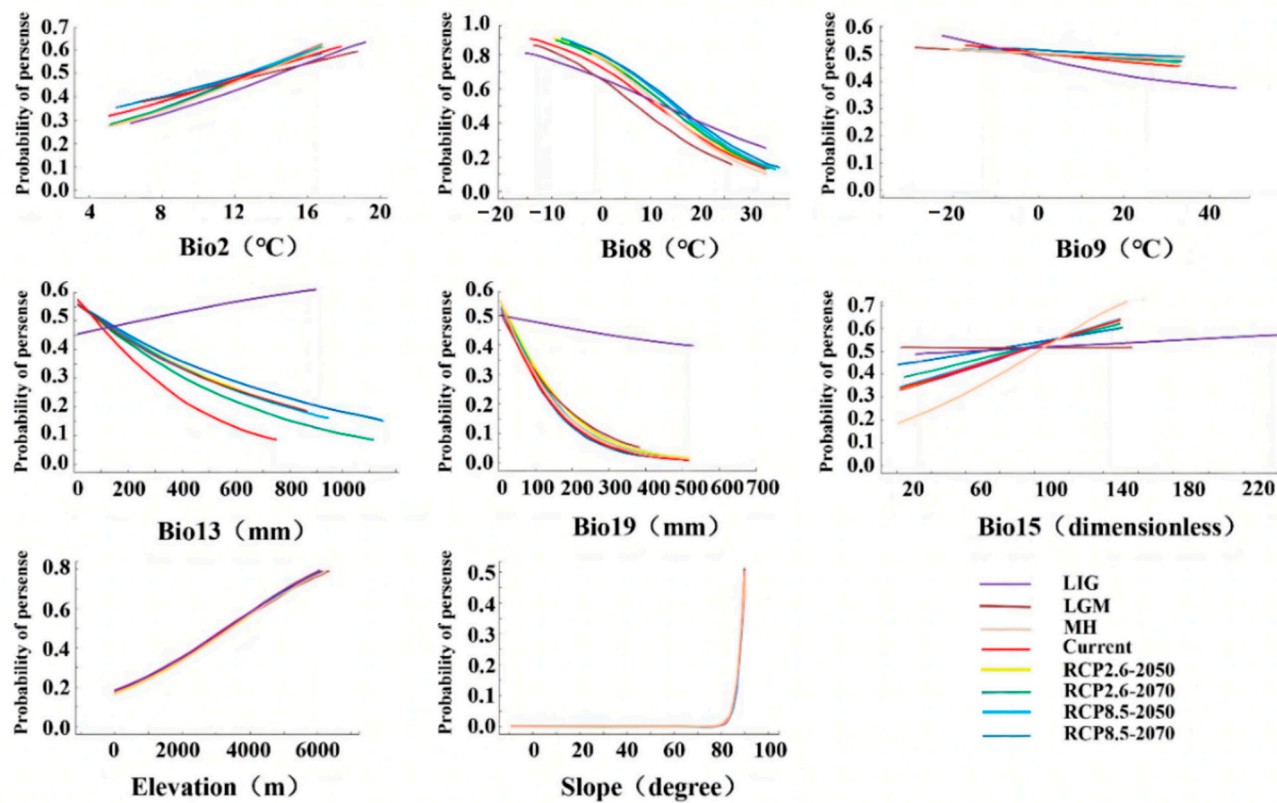

**Figure 6.** Response curves of records of nine environmental predictors used in the ecological niche model for *D. validus*.

### 3.4. Species Distribution Shifts under Different Climatic Scenarios

In general, limited expansion and contraction in the extent of highly suitable habitats were predicted across the suitable habitat area with climate change. The results are shown in Figures 7 and 8. MaxEnt projected that the area of suitable habitat losses would be mainly concentrated in the northern and eastern areas of the current habitat. Under the same climate scenarios, the range of appropriate habitats could increase in both the southern and eastern margins of the current suitable habitat, which is mainly in the Gangdise and Hengduan Mountains of China.

The SDMs of the LIG estimated that southern and eastern Tibet, western Sichuan, and southern Qinghai had more suitable habitats than the current distribution (Figure 7). However, the suitable habitats in central Tibet, southern and central Xinjiang, central and northern Qinghai, and western Sichuan in China contracted slightly. The MH model was similar to that of the LGM, with the estimated suitable habitat increasing mainly in the southern and eastern parts of the current habitat, the Gangdise and Hengduan Mountains of China (Figure 7). In future scenarios, MaxEnt estimated that the suitable habitat would increase mainly in the southern edge of the current habitat but decrease in the western parts (Figure 7). The suitable habitat area of *D. validus* in 2050 would be slightly larger than that in 2070 due to its higher expansion and lower contraction (Figure 8a).

The current distribution center for *D. validus* was predicted to be in northeastern Tibet (Figure 8b). However, during the LIG period, the habitat centroid was southwestern of the current centroid of suitable habitats. This centroid shifted to the northeastern region during the LGM. In the MH, the centroid was distant from its current location. Looking to a possible future under RCP2.6, the centroid would have shifted southeast by 2050 and then moved westward by 2070. The centroid was predicted to shift southward by 2050 and then westward by 2070, applying RCP8.5. In brief, the distribution core zone of suitable habitat would shift to the southwest in future climate scenarios.

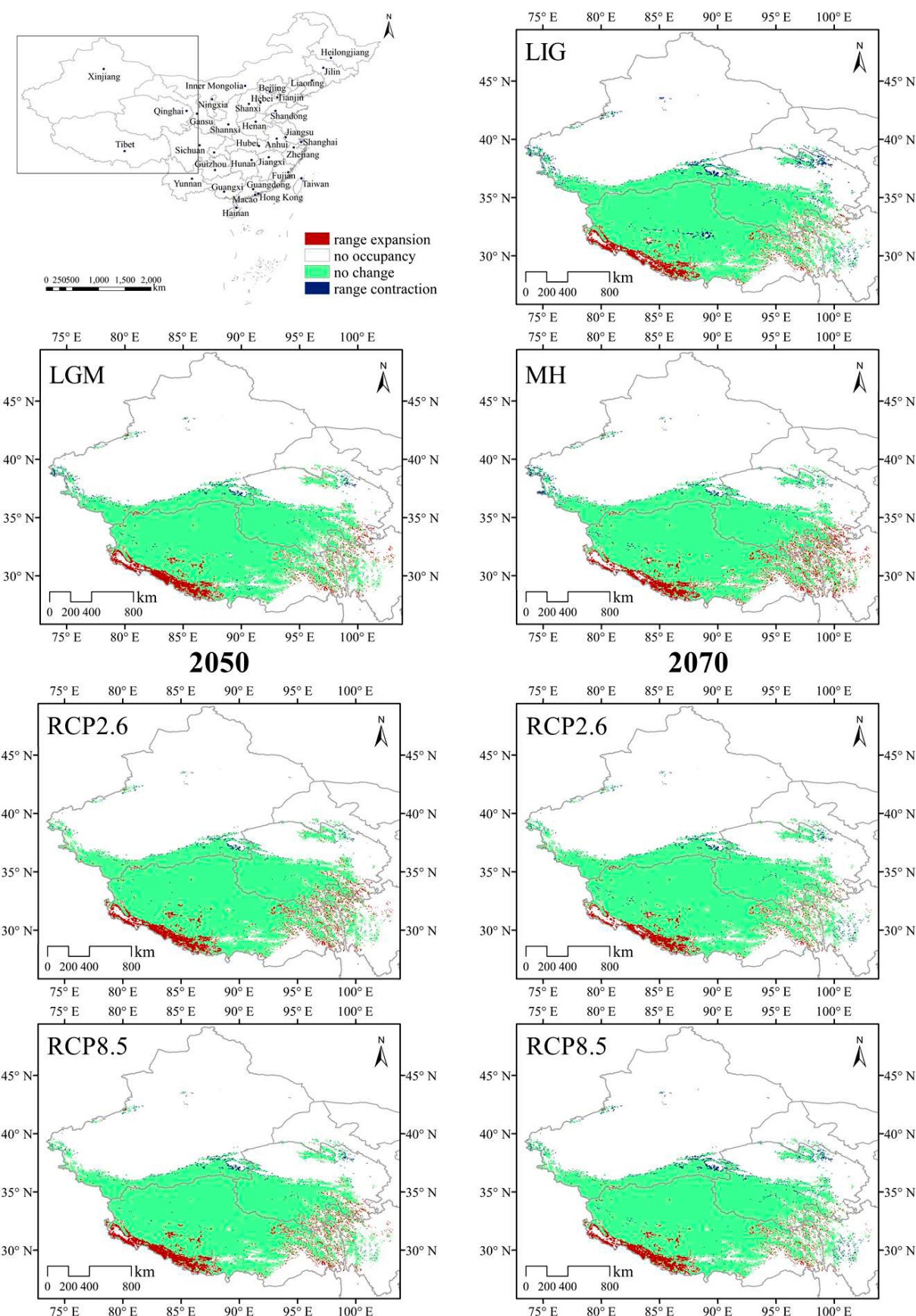

**Figure 7.** Species distribution models (SDMs) of *D. validus*. Models reflect habitat differences compared with the current model.

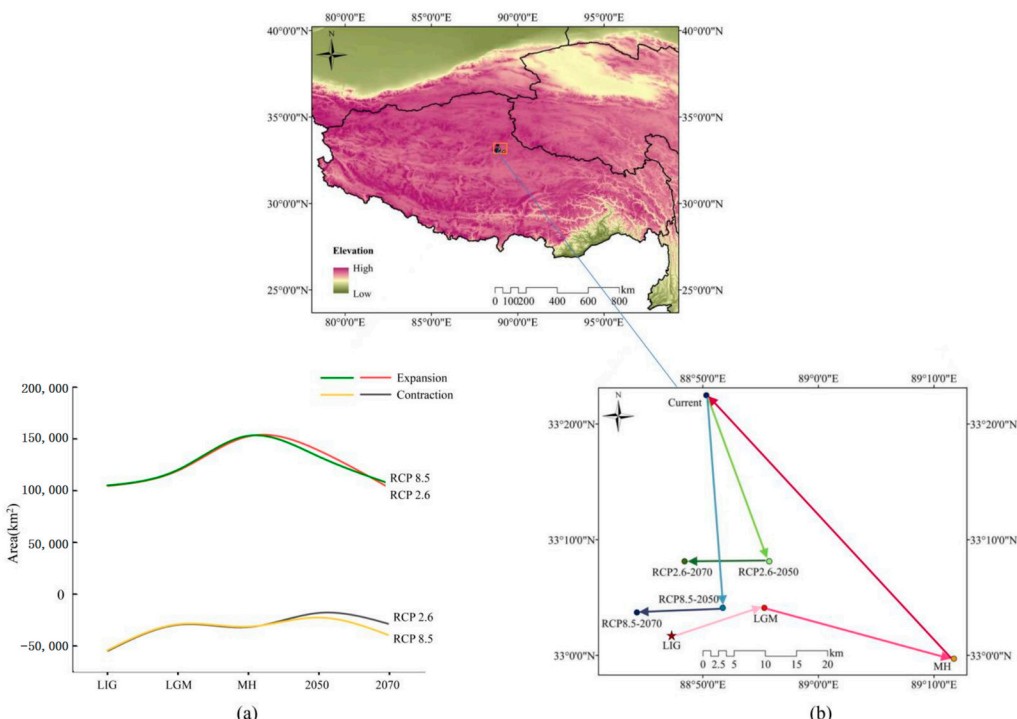

**Figure 8.** The distribution shifts of *D. validus* under different climate scenarios. (**a**) Dynamic changes in the distribution area of *D. validus* under four future and three past climate scenarios compared to the current suitable habitat area. (**b**) Core distribution shifts of *D. validus* under the four future and three past climate scenarios.

## 4. Discussion

Previously, *D. validus* was classified as an infraspecific taxon within *Didymodon rigidulus* Hedw due to their shared characteristics, such as leaf shape, curvature of the leaf margins, leaf areolation, and color of the laminal cells in reaction with KOH [32]. Our findings corroborate the observations made by Jiménez (2006) and Shuayib et al. (2017), indicating that the leaf cells of *D. validus* lacking papillae exhibit distinct characteristics compared to the middle and upper leaf cells of *D. rigidulus* with papillae (Figure 2). The structures of papillae are associated with water sparing and the avoidance of light radiation. The absence of warts on the surface of the leaf cells of this species, which occur in forest ecosystems, may be due to long-term adaptive evolution, exemplifying the integrity of plant morphology and function. The smooth nature of *D. validus* leaf cells facilitates water penetration by reducing surface tension, thereby enhancing water absorption. Moreover, this characteristic also promotes gas exchange, diffusion, and nutrient uptake, ultimately improving their adaptability in the forest edge ecosystem.

In addition, Kürschner and Neef (2012) highlighted that *D. validus* exhibits a pronounced capacity for asexual reproduction, as evidenced by its ability to generate numerous gemmae in leaf axils, thereby ensuring successful colonization and long-term habitat maintenance. However, the presence of gemmae as propagules in mosses is not always consistent despite their frequent consideration as a characteristic of species classification. Gemmae were present in *D. validus* from Europe and parts of Asia [19,53], while they were absent in China, according to Shuayib et al. (2017) and our own observations. This could mean that the structure of the gemmae will be affected more by long-term geographical isolation than by habitat conditions. However, we found that *D. validus* in dry lands had more branches and pseudoroots, dense mat-like clusters, and thicker cell walls than the wartless pair of mosses distributed in forest ecosystems. Persistent pressure from environmental stress may have facilitated plant adaptation under adverse conditions and the evolution of many survival mechanisms [54,55].

Although it is impossible to simulate the potential distribution of species with few sites, the prediction of the distribution pattern of *D. validus* is helpful for the discovery and conservation of narrow-range species in their corresponding habitats. Multi-model intercomparison studies have reported that the MaxEnt model, which is based on the maximum entropy principle, typically outperforms other SDMs in terms of high tolerance and high predictive accuracy, particularly for small sample sizes [56–58]. In this study, less distribution point information was used to predict the MaxEnt model, and the results showed that the cross-validation AUC of the *D. validus* model was 0.751 and the TSS value was 0.526, further supporting the suitability of our ArcGIS-based MaxEnt model. The simulation results are still reliable under the condition that the species distribution points are few, but uniform collection is guaranteed. In addition, the site information used in this study, the secondary data, and the obtained results showed that the distribution range of *D. validus* was narrow rather than unevenly investigated.

The MaxEnt model is frequently used by conservationists and managers to model endangered and invasive species in order to rehabilitate species while at the same time preserving their habitats [59,60]. It can utilize both continuous and categorical data and can incorporate interactions between different variables [61]. Moreover, the output of MaxEnt is continuous, allowing fine distinctions to be made between the modeled suitability of different areas, and these fine distinctions in predicted relative environmental suitability can be valuable to reserve planning algorithms [42]. However, because the dependence of the MaxEnt probability distribution on the distribution of occurrence locations is explicit, a difference between available occurrence records and background sampling may lead to inaccurate models that, in turn, may lead to inappropriate management decisions [62].

Jiménez (2006) documented *D. validus* growing in dense turf that was olive-green or brown-green. Its habitat is dominated by flaky soil, calcareous rocks, and dry talus at elevations from 200 to 1615 m. However, *D. validus* found in China was mainly distributed on soil over rocks and soil at the edge of the coniferous forest belt above an altitude of 2000 m, and the species showed a more suitable area and higher suitability with the increase in altitude. Our results highlighted a significantly larger suitable distribution range for *D. validus* than the current known distribution of the species in China (Figure 3). The potential range of *D. validus* is not limited to alpine forest ecosystems but may cover a wide range of humid regions and dry lands of the QTP. Shuayib (2018) pointed out that there are some morphological and structural differences in the growth performance of *D. validus* in different geographic regions that help it adapt to both arid and humid environments. Furthermore, by comparing the potential distribution patterns of *D. validus* under historical, current, and future climate scenarios, we observed that the highly suitable range of *D. validus* in China remains relatively stable under different climate conditions (Figure 4). We suspect that the high mountains of western China may provide a refuge for *D. validus* to cope with climate change. Many previous studies have shown that the QTP provides a refuge for many alpine plants to cope with vegetation migration events in different historical periods to reduce the influence of climatic events on their growth [21]. Refugia with complex topography may buffer against the impacts of climate change [63,64] and allow for the local persistence of species through successive periods of climate change [65]. This implies that the QTP provides a suitable habitat for *D. validus* even under changing climatic conditions.

In addition, our results showed that elevation and mean temperature in the wettest quarter were the predominant variables driving the potential distribution of *D. validus* (Figure 5). Previous studies have shown that altitude is usually a key eco-factor for the distribution of alpine species [66]. In mountainous areas, the elevational gradient causes changes in moisture, temperature, precipitation, and solar radiation [67]. Furthermore, the contribution of the mean temperature in the wettest quarter was significant. Temperature has been shown to affect the growth of bryophytes by determining their respiration rates [1]. Mild temperatures and relatively high water availability favor the continuous

production of sex organs [68]. In addition, fertilization time is dependent on the wettest period [69]. Most mosses also depend on moisture for growth and reproduction [70]. Therefore, we speculate that this is why *D. validus* is found in forest ecosystems. *Didymodon validus* may prefer a warm and humid season during the same period of rain and heat for development or reproduction. Precipitation in the coldest quarter was a constraint on the distribution of *D. validus* in our study. Ma and Sun (2018) suggested that precipitation will increase habitat suitability for bryophytes in the coldest season. In alpine regions, the coldest season is usually accompanied by low temperatures. Soil freezing may make water less readily available, while precipitation can provide plants with the needed moisture.

In our study, the response curves to the key climatic variables showed that *D. validus* exhibited a wider range of climatic adaptations during the LIG period relative to the other periods (Figure 6). The LIG period was characterized by a warm and humid climate and is considered one of the warmest periods globally for nearly 150 ka. This warm period may have led to the melting of snow and ice on the Tibetan Plateau and the expansion of lakes, providing favorable conditions for the growth and reproduction of *D. validus*. In addition, the habitat suitability of *D. validus* gradually stabilized after MH. During the LGM, mountain ranges were uplifted, and the climate became colder and drier, adversely affecting the survival of *D. validus*. Until the MH after the LGM, the climate warmed up, and the glaciers shrank in a large area [71]. These factors indicate that the adaptability of *D. validus* to climate in different periods is affected by climate change and that the warm and humid climate conditions during the LIG period provide opportunities for reproduction and survival.

The impact of climate change on the geographical distribution of plants is mainly reflected in changes in distribution range and area [72]. QTP is the highest alpine ecosystem in the world and one of the most sensitive to climate change [21]. Numerous species are confronted with the shrinking of their spatial distribution in the case of continuous warming on the plateau or even the possibility of extinction. However, under such a long time and spatial variation in this study, *D. validus* has managed to settle successfully and steadily in a high-altitude environment. This not only demonstrates the ecological adaptability of this species, which is capable of withstanding adverse conditions on the plateau, but also indicates that, despite its limited distribution in China, *D. validus* does not retreat from its original habitat with the changing climate. Compared with other narrow-ranged species, there is no need for excessive habitat protection or concern for this species.

According to previous studies, the differentiation of *D. validus* was completed in the Miocene [34]. Our MaxEnt model reconstructed that the simulated distribution range of *D. validus* was stable and did not shrink during the LIG, LGM, and MH and that the area of expansion had been increasing (Figure 7). Quaternary glaciation has always been regarded as an important event in the study of phylogeography, and drastic fluctuation in the climate is generated by the recurrence of glaciation [73]. Many studies have shown that the QTP never formed a uniform ice sheet during the glacial period, but its complex topography made it possible to have relatively warm and moist places for organisms to survive, even in harsh climatic conditions [74]. From the LIG to the LGM, concomitant with climate cooling, the habitat suitable for *D. validus* is still expanding along the southern and eastern margins of the QTP, albeit at a slower rate. In the LGM of the Quaternary, the temperature of the Asian continent was approximately 5–11 °C lower than it is currently, and the climate warmed after the LGM [75]. At the MH, the expansion area of potential habitats for *D. validus* was substantially increased compared to the other periods, but these were mostly included within the current potential range. Many alpine plants adapt to warming by moving upward or northward in altitude, and the ultimate consequence can be extinction, a phenomenon often described as "nowhere to go" [21]. Furthermore, narrow-range species usually live under very specific environmental conditions and are the most vulnerable to climate change [76]. Some studies have shown that not only are they more likely to face distribution changes and range shrinkage, but they might

also disappear due to the strong alteration of their climatic envelope throughout their entire region [77]. Despite the predicted low suitability of climate conditions, it is possible that species persist under changing conditions through adaptation or plasticity [78,79]. Our findings suggest that the distribution change in *D. validus* fits with the latter. In addition, through the observation of the species under long-term climate and spatial changes, it was found that although the center of the suitable habitat of *D.validus* showed a trend of migrating to the southwest in the future, there was no significant change in longitude and latitude, and the migration distance was relatively short (Figure 8). This may be due to the limitations of geographical isolation. And this short-distance migration may be caused by climate fluctuations in a certain period of time. We underestimated the adaptability of the moss.

### 5. Conclusions

The present study employed the MaxEnt model to reveal that the suitable habitat of *D. validus* as a narrow-range moss species is not limited to alpine forest ecosystems. In fact, *D. validus* exhibits cold and drought resistance on the QTP of China. The distribution pattern of *D. validus* is mainly defined by key environmental variables, including altitude and mean temperature, in the wettest quarter. In general, despite dramatic climatic and spatial changes over a long period of time, the range of *D. validus* has not been radically altered, nor has it shown a significant tendency to shrink. However, due to the complex terrain of the QTP, it provides refuge and mitigates the adverse effects of climate change. Meanwhile, *D. validus* showed unique adaptability in China, enabling it to cope with the challenges of adverse climatic conditions. In the context of future climate warming, the suitable habitat center of *D. validus* may move southwestward, but it will not completely withdraw from the distribution range of China, and its potential distribution range will remain relatively stable. This study provides an important reference for addressing the impact of climate change on alpine narrow-range moss species and provides strong support for future distribution and conservation strategies.

**Author Contributions:** Conceptualization, J.K. and B.Z.; methodology, T.W.; software, T.W.; validation, C.P., T.B. and Q.W.; formal analysis, T.W., C.P. and J.K.; investigation, J.K.; resources, J.K. and B.Z.; writing—original draft preparation, T.W., C.P. and J.K.; writing—review and editing, J.K. and B.Z.; visualization, T.W., T.B. and Q.W.; supervision, J.K. and B.Z. All authors have read and agreed to the published version of the manuscript.

**Funding:** This work was supported by the National Natural Science Foundation of China (grant No. 42001045, 32060051) and Shenzhen Key Laboratory of Southern Subtropical Plant Diversity (grant No. 99203030).

**Data Availability Statement:** The data that support the findings of this study are available from the corresponding author upon reasonable request. Restrictions apply to the availability of these data, which were used under license for this study.

**Acknowledgments:** The authors are grateful to the regional editors and anonymous reviewers whose invaluable insights have played a pivotal role in enhancing the quality of this manuscript.

**Conflicts of Interest:** The authors declare no conflict of interest.

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
