# Peer review of "Response of a Sylvan Moss Species (Didymodon validus Limpr.) with a Narrow Distribution Range to Climate Change"

_forests, doi:10.3390/f14112227_

Round 1
Reviewer 1 Report
Comments and Suggestions for Authors
I am most bothered by possible taxonomic problems. The authors say that the Chinese material grows in a different habitat than the European (type) material, and that European material has gemmae and the Chinese does not. Are we really sure that we are dealing with the same species in Europe and China? I'd like something that gives me confidence in this.
I think the methodology is great, but I'm concerned if they are really dealing with Didymodon validus. Also, in looking at the maps, for example in Fig. 3, I can't really see much difference between the different maps. Perhaps the differences could be made more prominent.
Comments on the Quality of English LanguageOverall, the English is pretty good. I would suggest that the word "sylvatic" in the title, which is an odd word in English, be changed to "sylvan."
Reviewer 2 Report
Comments and Suggestions for Authors
The manuscript entitled “Response of a sylvatic moss species (Didymodon validus Limpr.) with a narrow distribution range to climate change" used the Maxent distribution modeling approach together with relevant environmental variables to construct the past and predict the current and future distribution patterns of Didymodon validus. The manuscript is well written, and the drawn conclusions are coherent with the obtained results. Although similar methodologies are common, the results of the study could have useful implications for management actions. The manuscript requires some changes before it's ready for publication.
Title:
Title does not need a full stop.
Abstract:
Line 11: How can the past be predicted? Please refine this expression; one can reconstruct the past.
Please outline the aim of the study clearly.
Introduction
Line 56-66: I suggest expanding on what exactly previous studies have done and justifying how this study is different from them. In this context, the introduction could benefit from an extra review of the target species on a regional or global scale. Please review previous studies and mention the techniques used in studying the target species.
Material and Methods
2.2. Study area and species occurrence
Line 98: Which sampling technique was used? Please clarify.
- It is not clear whether any spatial filtering was performed for the occurrence points.
- It is not clear how many occurrence points were collected.
“Fig. 1” please revisit this map; the “occurance” should be “occurrence”
2.3. Environmental variables and climate change scenarios
Line 117: "RCP" Why SSP was not used? According to the “worldclim” website, the Representative Concentration Pathways (RCPs) are now obsolete, and they recommend the use of the new version (CMIP6 data), i.e., the Shared Socio-economic Pathways (SSPs) in similar studies.
2.4. Distribution modeling
Line 141: “maximum number of background points (10,000)” Why were 10,000 points used?
Again, how many occurrence points were used for building the models? Please clarify.
Line 148: “0.2” Please justify why this figure was selected as a threshold.
Discussion
-A small section highlighting the benefits of the applied modeling techniques in establishing priority zones for management actions is necessary. I suggest:
- In addition, the limitations of the applied modeling techniques in a few sentences should be added, please.
Conclusions
Line 439-440: “The present study used the MaxEnt model to explore whether D. validus is not a narrow-range moss species in” Please revisit this sentence. It is not clear.
Line 144: “The distribution pattern of D. validus is mainly defined by key climatic variables, including altitude and” Altitude is not a climatic variable. Please revisit the entire conclusion.
Round 2
Reviewer 1 Report
Comments and Suggestions for Authors
The maps seem much improved.
Comments on the Quality of English LanguageOverall, the English seems good.
Reviewer 2 Report
Comments and Suggestions for Authors
The manuscript is sufficiently improved.